# Three-Dimensional Modelling of Indexed Papillary Muscle Displacement in Patients Requiring Mitral Valve Surgery Using Four-Dimensional Echocardiography Variables

**DOI:** 10.3390/jcm13247503

**Published:** 2024-12-10

**Authors:** Zhi Xian Ong, Ashlynn Ai Li Ler, Liang Shen, Theo Kofidis, Lian-Kah Ti, Faizus Sazzad

**Affiliations:** 1Department of Surgery, Yong Loo Lin School of Medicine, National University of Singapore, Singapore 117599, Singaporesurtk@nus.edu.sg (T.K.);; 2Department of Cardiac, Thoracic and Vascular Surgery, National University Heart Centre, Singapore, Singapore 119074, Singapore; 3Biostatistics Unit (BSU), Department of Medicine, National University of Singapore, Singapore 119228, Singapore; medsl@nus.edu.sg; 4Department of Anaesthesiology, National University Hospital, National University Heart Centre, Singapore, Singapore 119074, Singapore

**Keywords:** echocardiographic imaging, ischemic mitral regurgitation, degenerative mitral regurgitation, papillary muscle displacement, indexed displacement, mitral valve

## Abstract

**Background**: Two-dimensional and three-dimensional echocardiographic imaging are commonly used in assessing ischemic mitral regurgitation (IMR) and degenerative mitral regurgitation (DMR) in patients with mitral valve disease. However, the use of 4D echocardiographic imaging has not yet been reported. The objectives of this study were to explore the efficacy of utilizing 4D echocardiographic variables, determine papillary muscle displacement in patients with either IMR or DMR, and compare the differences in papillary muscle displacement between groups. **Methods**: Thirty-four patients were divided into two groups: Group 1 (with IMR) and Group 2 (with DMR). Using clinical ultrasound software, 4D echocardiographic variables were obtained and compared between the groups. Pearson’s product–moment correlation test was used to assess the relationship between the presence of IMR and both papillary muscle displacement and indexed papillary muscle displacement. **Results**: The mean values for papillary muscle displacement in Groups 1 and 2 were 38 ± 6.7 mm and 31.8 ± 6.1 mm, respectively. Indexed papillary muscle displacement was 22.8 ± 3.7 mm in Group 1 and 18.4 ± 3.5 mm in Group 2. There were statistically significant correlations between the presence of IMR and papillary muscle displacement (*p* = 0.009) and indexed papillary muscle displacement (*p* = 0.002). A significant correlation was also observed between IMR and PL (*p* = 0.001), with mean values of 15.7 ± 3.9 mm in Group 1 and 20.2 ± 5.6 mm in Group 2. **Conclusions**: Four-dimensional echocardiography is effective in evaluating morphological variations in IMR. It successfully determined papillary muscle displacement in patients undergoing mitral valve surgery and demonstrated a positive correlation between IMR and indexed papillary muscle displacement.

## 1. Introduction

The most common cardiac imaging method used today is 2D (two-dimensional) echocardiography, which utilizes Doppler sound waves to generate images of the heart’s internal anatomy. Despite its frequent use, several limitations are associated with this technique. First, the calculations for ejection fraction and left ventricular volume rely on assumptions that may not hold in dilated hearts [1,2]. Second, the lack of standardized interpretation methods for 2D echocardiography often results in observer variability, complicating clinical diagnosis. Lastly, the 2D nature of the images makes it difficult to precisely locate cardiac abnormalities [3]. In response, 3D (three-dimensional) echocardiography was developed, allowing for a more comprehensive analysis of the heart by compiling continuous 2D slices into 3D images. Real-time 3D echocardiography (RT3DE), also known as 4D (four-dimensional) echocardiography, eliminates the need for ECG and respiratory cycle gating, which was required in earlier 3D techniques [4]. It provides real-time 3D visualization of cardiac anatomy using a transthoracic or transesophageal matrix array probe [4], enabling surgeons to assess valve abnormalities, detect wall motion defects, and pinpoint cardiac issues more accurately [3].

Recent studies have compared 4D echocardiography with traditional methods like cardiac magnetic resonance imaging (CMR) and radionuclide ventriculography [5,6,7,8,9], demonstrating the superiority of 4D echocardiography in terms of measurement reproducibility [7,8,10]. Some studies even found that 4D echocardiography offers similar reproducibility to CMR, the current gold standard for non-invasive cardiac imaging [11]. Ischemic mitral regurgitation (IMR) results from changes in left ventricular structure due to ischemic heart disease, leading to increased papillary muscle displacement, mitral leaflet tethering, and restricted closure [12]. Degenerative mitral regurgitation (DMR) results from structural degeneration of the mitral valve leaflets or chordae tendineae. Previous studies have used 3D echocardiography to assess mitral valve annuloplasty and geometric abnormalities in IMR and DMR [13,14]. However, there is limited research on using 4D echocardiography to measure papillary muscle displacement in these patients [15].

Despite the extensive research on 3D echocardiography in IMR and DMR, no studies have yet explored the use of 4D echocardiography in determining papillary muscle displacement in these patients. This study aims to address that gap by utilizing transesophageal 4D echocardiographic measurements to evaluate papillary muscle displacement in patients with IMR and DMR pathologies. Through comparison of two cohorts—patients with and without IMR—we aim to assess the role of 4D echocardiography in evaluating papillary muscle displacement in IMR and DMR.

## 2. Materials and Methods

A retrospective cohort study was conducted in the Department of Surgery at Yong Loo Lin School of Medicine, National University of Singapore, from 1 January 2017 to 1 April 2019. Data were collected from 142 patients who underwent transesophageal 4D echocardiographic assessment at National University Hospital, Singapore (NUH). Patients were excluded if they had congenital mitral valve anomalies of heterogeneous pathologies, secondary mitral regurgitation (non-ischemic MR), other valvular heart diseases, prior cardiac surgery, no 4D echocardiography performed, or missing data. After applying the exclusion criteria, 34 patients remained and were divided into two groups based on the presence of IMR and DMR (Appendix A).

### 2.1. Data Collection

A list of variables was prepared, and a data collection form was created prior to gathering the patients’ echocardiographic data. In collaboration with the Department of Anaesthesiology at NUH, data from 34 patients were collected, and their echocardiographic data was stored on compact discs (CDs). Demographic data including gender, age, height, weight, body mass index (BMI), body surface area (BSA), diagnosis, and the presence of IMR were recorded.

The echo data from the CDs were subsequently read and analyzed. The variables extracted included commissure to commissure (CC), trigone to trigone (TT), annulus height (A_2_P_2_), angle of saddle tilt, calculated mitral valve area, perimeter projection, anterior leaflet (AL), posterior leaflet (PL), coaptation lengths (anterior and posterior), chord lengths (CH-A and CH-P) in systole and diastole, inter-papillary distance, number of papillary muscle heads (anterolateral and posteromedial), left ventricular end-diastolic and end-systolic internal diameters (LVIDd and LVIDs), ejection fraction (EF), maximum pressure gradient (P_max_), and mean pressure gradient (P_mean_).

### 2.2. QLAB 13 Software Analysis

The software used to analyze the 4D echocardiography data was QLAB 13, an experimental, unreleased software developed by PHILIPS and TomTec, which was approved by the FDA on 18 June 2019. A set of Q-apps analysis tools equipped in the software could be selected for the quantitative analysis of echocardiographic data. These tools included 4D Mitral Valve Assessment (MVA), Mitral Valve Navigation (MVN), General Imaging 3DQ (GI3DQ), etc. The MVA feature was used to perform echo analysis, allowing the extraction of essential variables for the study. The positions of the mitral valve annulus and leaflets were automatically generated by the software, outlined using “points” and “lines” superimposed on the approximated margins of the mitral valve, as shown in Figure 1. These “points” and “lines” could be adjusted manually to gain a more accurate approximation of the actual position of the mitral valve using the next three stages of review, namely, view adjustment, static model (Figure 2), and dynamic model (Figure 3) reviews.

In addition to this, QLAB 13 also had the ability to perform quantitative analysis on a single frame throughout the cardiac cycle to obtain other parameters such as Ch-A, Ch-P, and inter-papillary distance. Mitral valve measurements were obtained after completing these stages (Figure 4). The results were exported in .txt format and arranged into measurement groups: “Annulus”, “Leaflets”, “Coaptation”, “Miscellaneous”, “Manual Measurements”, and “Dynamic Measurements”. These measurements were then extracted and compiled into an Excel sheet, summarizing the data for all 34 patients involved in the study. After collecting the 4D echocardiographic variables of all the patients, the data were compiled and edited meticulously and anonymously. The data were also checked again for any missing values or discrepancies. All omissions and inconsistencies were corrected and removed methodically. Additionally, other variables not derived directly from the software, such as indexed papillary muscle displacement, were manually calculated using the following formula: Indexed papillary muscle displacement = Inter-papillary distance/Body surface area. These values were then included in the data collection form.

### 2.3. 3D Modelling

To visualize papillary muscle displacement from a healthy heart to one with IMR, a 3D animation was created using Blender [16]. The animation depicted a beating heart transitioning from diastole to systole within one cardiac cycle, rotating the heart through 270°.

### 2.4. Statistical Analysis

All statistical analyses were performed using RStudio [17]. A generalized structural equation model was applied for data analysis, and all statistical tests were performed with a 95% confidence interval (equivalent to a *p*-value of 0.05 in a two-sided test). Measurements were presented as mean ± standard deviation and *p*-values. Pearson’s product–moment correlation test was used to assess the strength of the linear relationship between the variables, with significance determined using the cor.test function. For all computations, the significance level was set at *p* < 0.05.

## 3. Results

### 3.1. Demographic and Clinical Characteristics

The demographic data of the patients in Groups 1 and 2 are shown in Table 1. The overall study population included patients aged between 39 and 81 years. The mean age of this population was 65.0 ± 7.3 years in Group 1 and 57.6 ± 11.7 years in Group 2. The majority of the patients were in the age group of 50 to 70 years. There was no significant correlation in age between the two groups (*p* = 0.058). There were 22 males (64.7%) and 12 females (35.3%) in total. The mean sex ratio in Group 1 was 1.3 ± 0.5, with 26.5% males and 8.8% females, while the mean sex ratio in Group 2 was 1.4 ± 0.5, with 38.2% males and 26.5% females. There was no significant correlation in sex between the two groups (*p* = 0.369).

Thirdly, the mean weights in Groups 1 and 2 were 61.7 ± 8.2 kg and 67.8 ± 14.8 kg, respectively. There was no significant correlation in weight between the two groups (*p* = 0.196). The mean height in Group 1 was 1.63 ± 0.1 m, whereas the mean height in Group 2 was 1.6 ± 0.1 m. There was no significant correlation in height between the two groups (*p* = 0.986). Group 2 had a higher mean BMI of 25.4 ± 4.0 kg/m^2^ compared to Group 1, which had a mean BMI of 23.3 ± 3.4 kg/m^2^. There was no significant correlation in BMI between the two groups (*p* = 0.136). Lastly, the mean BSA in Group 1 was 1.7 ± 0.1 m^2^, and in Group 2, it was 1.7 ± 0.2 m^2^. There was also no significant correlation between the two groups in BSA (*p* = 0.301).

### 3.2. Surrounding Structures and Mitral Leaflet Measurements

Table 2 shows the results of the surrounding structures of the patients’ left ventricle and the results of the patients’ mitral leaflets, respectively. There were statistically significant correlations in inter-papillary distance, indexed papillary muscle displacement, and posterior leaflet height between the two groups. For Group 1, the mean inter-papillary distance was 38.0 ± 6.7 mm, while for Group 2, the mean value was 31.8 ± 6.1 mm. The mean indexed papillary muscle displacements were 22.8 ± 3.7 mm and 18.4 ± 3.5 mm in Groups 1 and 2, respectively. As shown in Table 2, both inter-papillary distance (*p* = 0.009) and indexed papillary muscle displacement (*p* = 0.002) showed significant correlations between the two groups and the presence of IMR. The mean values for the posterior leaflets were 15.7 ± 3.9 mm and 20.2 ± 5.6 mm for Groups 1 and 2, respectively. As seen in Table 2, the *p*-value for the posterior leaflets was <0.05 (*p* = 0.018), suggesting a significant correlation between the height of the posterior leaflets and the presence of IMR.

### 3.3. Mitral Annulus Measurements

Table 2 demonstrates the mitral annulus parameters of the study population. Firstly, the mean CC in Group 1 was 37.9 ± 6.8 mm, while the mean CC in Group 2 was 38.0 ± 8.9 mm. From the calculated *p*-value (*p* = 0.978), it was observed that there was no significant correlation in CC between the two groups. Secondly, the mean TT in Groups 1 and 2 was 24.2 ± 3.4 mm and 22.6 ± 6.5 mm, respectively. As the *p*-value was > 0.05 (*p* = 0.438), the correlation between the two groups in TT was not significant. The mean A_2_P_2_ in Group 1 was 9.0 ± 2.3 mm, whereas the mean A_2_P_2_ in Group 2 was 8.0 ± 2.9 mm. There was no statistically significant correlation in A_2_P_2_ between the two groups (*p* = 0.293). The mean angle of tilt of the saddle in Group 1 was 125.1 ± 10.3°, while in Group 2, the mean angle was 121.7 ± 18.4°. Based on the calculated *p*-value (*p* = 0.556), we determined that there was no significant correlation between the two groups in terms of the angle of tilt of the saddle. Group 1 had a mean area of 12.4 ± 4.5 cm^2^, while Group 2 had a mean area of 13.2 ± 5.9 cm^2^. As the *p*-value was 0.664, there was no significant correlation between the two groups. Lastly, the mean perimeter was 13.0 ± 2.1 cm in Group 1, while in Group 2, the mean perimeter was 13.1 ± 3.0 cm. As the *p*-value was >0.05 (*p* = 0.889), the correlation in terms of the perimeter between the two groups was not significant.

### 3.4. Chordae Tendineae Measurements

Table 2 shows the mean, standard deviation, and *p*-value of the chordae tendineae in both systole and diastole. The mean for Ch-A (s) was 31.0 ± 5.4 and 32.3 ± 6.9 for Group 1 and Group 2, respectively. Since the *p*-value was >0.05 (*p* = 0.597), there was no significant correlation in Ch-A (s) between the two groups. Secondly, the mean of Ch-P (s) was 25.3 ± 4.6 and 26.7 ± 6.9 for Group 1 and Group 2, respectively. As *p* = 0.549, there was no significant correlation between the two groups in terms of Ch-P (s). The mean for Ch-A (d) was 27.5 ± 7.8 and 27.8 ± 7.1 for Group 1 and Group 2, respectively. From the calculated *p*-value (*p* = 0.918), there was no significant correlation between the two groups for Ch-A (d). Lastly, the mean of Ch-P (d) was 25 ± 7.3 for Group 1 and 25.7 ± 6.0 for Group 2. Based on the calculated *p*-value (*p* = 0.802), the correlation between the two groups was also not significant.

### 3.5. Papillary Muscle Measurements

Table 2 shows the results of the papillary muscles in both groups. As all the patients included in the study had one anterolateral and one posteromedial papillary muscle head, the mean for both groups was 1.0 ± 0.0, indicating that there was no significant correlation between the papillary muscle and the presence of mitral regurgitation.

### 3.6. Left Ventricular Dimensions

Table 2 shows the results of the different dimensions for Groups 1 and 2. The mean LVIDd of Group 1 was 5.3 ± 0.7 cm, while the mean LVIDd of Group 2 was 5.0 ± 0.7 cm. Since *p* = 0.186, which is >0.05, there was no significant correlation, indicating that LVIDd was not significantly correlated with the presence of IMR. The mean LVIDs were 3.8 ± 0.9 cm and 3.5 ± 0.7 cm for Groups 1 and 2, respectively. With *p* = 0.242, which is >0.05, there was no significant correlation between LVIDs and the presence of IMR. The mean ejection fraction (EF) for Group 1 was 47.9 ± 12.3%, and for Group 2, it was 54 ± 9.2%. Since the *p*-value was >0.05 (*p* = 0.113), there was no significant correlation between ejection fraction and the presence of IMR.

### 3.7. Baseline Hemodynamic Profile of the Study Population

Table 2 shows the results of the baseline hemodynamic profile in both groups. In Group 1, the mean P_max_ was 18.1 ± 26.9 mmHg, while in Group 2, the mean P_max_ was 16.8 ± 28.3 mmHg. There was no significant correlation between P_max_ and the presence of IMR (*p* = 0.900). The mean P_mean_ was 10.4 ± 18.2 mmHg for Group 1, while the mean P_mean_ was 9.7 ± 18.1 mmHg in Group 2. As the *p*-value was >0.05 (*p* = 0.911), there was no significant correlation between both variables. Hence, this indicates that there was no correlation between the baseline hemodynamic profile and the presence of IMR.

## 4. Discussion

To our knowledge, this is the first study to date that utilizes 4D echocardiography to examine the indexed papillary muscle displacement in patients with a variety of mitral valve pathologies, including IMR and DMR. As mentioned previously, the software used was QLAB 13, which was still in its developmental stage at the time of our study. Despite this, we determined that the software demonstrated significant promise in generating a 4D visualization of the heart while providing accurate measurements of the inter-papillary distance and other variables related to the structure and function of the mitral valve.

From our results, we observed that the inter-papillary distance, indexed papillary muscle displacement, and posterior leaflet length were larger in patients with IMR. This could be due to the remodeling of the left ventricle following ischemia in patients with ischemic heart disease, causing the papillary muscles to move farther apart and the posterior leaflet to expand as changes occur in the morphology of the dilating heart.

Our results were further supported by several articles indicating a high correlation between mitral insufficiency and papillary muscle dysfunction [18,19,20,21]. Robert et al. stated that the papillary muscle contracts abnormally due to ischemia or infarction, weakening the systolic support for the valve leaflets and eventually leading to their prolapse into the left atrium [18]. Additionally, mitral leaflet closure is restricted due to ventricular changes. Left ventricular dilation is commonly associated with papillary muscle dysfunction, as described by Mittal et al. and Madu et al. [19,20]. The resultant dyskinetic wall motion in the area of a papillary muscle causes the muscles to lose their normal structural support, pulling them away from the valve orifice [19]. The greater the distance between the papillary muscles, the more the leaflets can stretch [21]. This increases tension on the leaflets, leading to incomplete mitral leaflet closure and subsequent mitral insufficiency [19,20].

Furthermore, the association between mitral insufficiency and papillary muscle dysfunction is supported by Chaim’s study, where myocardial infarction was deliberately induced in various regions of dog hearts using “ameroid constrictors” to evaluate the relationship between papillary muscle infarction, heart dilation, and the development of mitral insufficiency. The results demonstrated that papillary muscle infarction alone was not sufficient to produce mitral insufficiency. Instead, dilation or dyskinesia, combined with targeted papillary muscle paresis, likely leads to malalignment or a faulty foundation for the papillary muscles, contributing to their malfunction and, consequently, mitral valve incompetency [21].

Our research demonstrates that 4D echocardiography is useful in determining the inter-papillary distance in patients with IMR and DMR. Furthermore, understanding the increased papillary muscle displacement in patients with IMR could help clinicians and surgeons reduce this displacement to the appropriate extent while attempting to repair the mitral valve.

We recognize the importance of comparing the results from this study using 4D echocardiography with those from other studies employing 3D echocardiography and cardiac resonance imaging (MRI). Such comparisons would allow us to validate the findings from 4D echocardiography, as each imaging technique has its own strengths and limitations. Cross-validation with more established methods like 3D echocardiography and MRI would help ensure the reliability and accuracy of the results. Additionally, different imaging modalities may vary in clinical applications, resolution, and accuracy. However, given the retrospective nature of this study, a direct comparison between the modalities was beyond the scope of our analysis. The homogeneity of data across different techniques could not be guaranteed, and as such, we are unable to make such comparisons.

Looking ahead, 4D echocardiography holds significant promise for a wide range of clinical applications beyond its current use in IMR. This technique can be particularly beneficial in the assessment and management of other cardiac conditions such as DMR, congenital heart defects, and valvular heart diseases. Its ability to provide dynamic, real-time visualization of heart function and valve motion can enhance preoperative planning, guide surgical interventions, and improve patient outcomes. Moreover, 4D echocardiography could be applied in monitoring the progression of heart failure, evaluating left ventricular function, and assessing post-surgical valve repair or replacement. As technology advances, the potential for 4D echocardiography to aid in the personalized management of various cardiac conditions will continue to grow, making it a powerful tool in both clinical practice and research.

### Limitations

The study is retrospective in nature, which means it is subject to the inherent limitations associated with retrospective analyses. It reflects the experiences of a single center, so the results should be generalized with caution. Variability in the operator’s technique for obtaining echocardiographic images and inter-operator variability could introduce inherent bias; however, this was meticulously addressed in this series through repeated training and evaluations. The patient groups were analyzed without subdividing them based on the number of images captured or utilizing the new software to analyze 4D variables, which could have enhanced the study’s results. This limitation was partly due to the relatively small number of cases performed in this series, as the new technology was only recently introduced at the National University Heart Center, Singapore.

## 5. Conclusions

Four-dimensional echocardiography is a valuable tool for measuring morphological variations in the mitral valve and has been successfully used to demonstrate positive correlations between the presence of IMR and factors such as papillary muscle displacement, the height of the posterior leaflet, and indexed papillary muscle displacement.

## Figures and Tables

**Figure 1 jcm-13-07503-f001:**
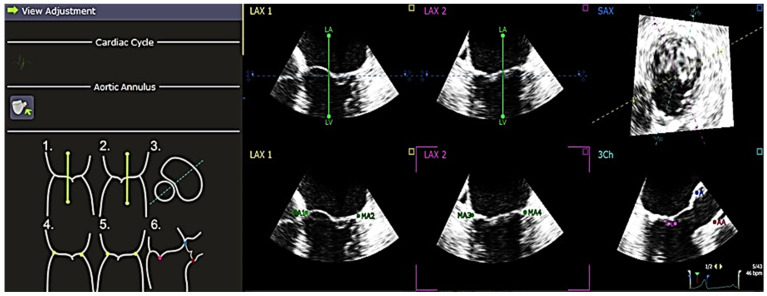
Alignment of “points” and “lines” to mitral valve—view of the mitral valve in the QLAB 13R software. The line in green marks the position of the left atrium and the left ventricle. The points demarcate the position of the mitral annulus, aortic annulus, anterior point, and coaptation/valve closure. There is a total of 7 points and 2 lines under “View Adjustment”, which need to be adjusted according to the reference provided at the left side. During the adjustment, it is allowed to pause at a single frame, which is helpful for placing the points and lines in their respective positions more accurately. After finishing the “View Adjustment”, it proceeds to “Static Model Review”. *A* = *anterior point*, *AA* = *aortic annulus*, *CL* = *coaptation/valve closure*, *MA* = *mitral annulus*, *LA* = *left atrium*, *LV* = *left ventricle*, *LAX* = *long axis*, *SAX* = *short axis*.

**Figure 2 jcm-13-07503-f002:**
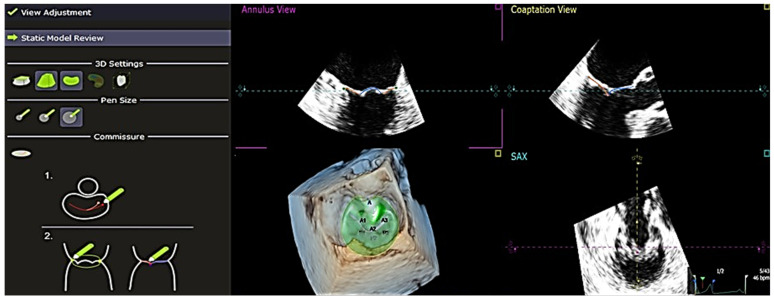
“Static Model Review” shows a static model of the mitral valve generated using QLAB 13R software showing the anterior and posterior leaflets. The lines are adjusted according to the reference found at the left under the “Static Model Review”. There are two views provided, which are annulus and coaptation views. For the two views, it is important to ensure that the line is aligned with the mitral valve. *A1*, *A2*, *A3* = *anterior leaflet*, *P1*, *P2*, *P3* = *posterior leaflet, SAX* = *short axis*.

**Figure 3 jcm-13-07503-f003:**
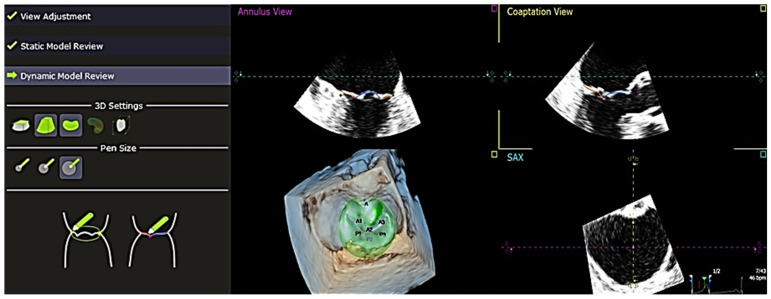
“Dynamic Model Review” shows alignment in the generated dynamic model review—this provides the visualization of both annulus and coaptation once the “Static Model Review” is completed. Both the “Static Model Review” and “Dynamic Model Review” play an important role in the visualization of the valvular complex and its geometric properties. *A1*, *A2*, *A3* = *anterior leaflet*, *P1*, *P2*, *P3* = *posterior leaflet*, *SAX* = *short axis*.

**Figure 4 jcm-13-07503-f004:**
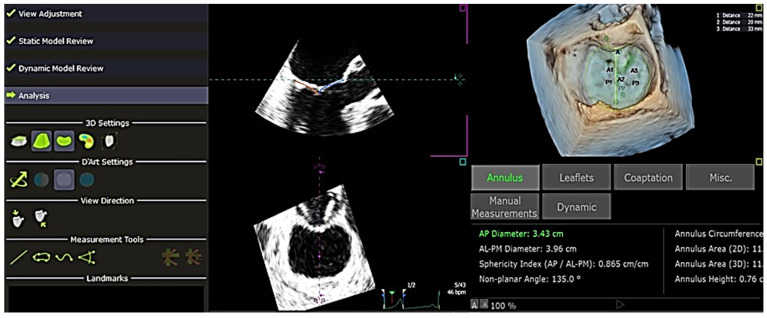
Analysis of the final outcome of the generated measurements of the mitral valve—at the end of each analysis, complete measurements are automatically calculated. These are height of the leaflets and coaptation length, anteroposterior diameter, anterolateral–posteromedial diameter, sphericity index, non-planar angle, annulus circumference, annulus area (2D), annulus area (3D), and annulus height of the mitral valve estimated using the QLAB 13R software. The measurements can be exported in 2 types of files: DICOM SR and .txt format. *AL-PM* = *anterolateral-posteromedial*, *AP* = *anteroposterior*.

**Table 1 jcm-13-07503-t001:** Demographic data of the patients include the presence of ischemic mitral regurgitation (IMR) and degenerative mitral regurgitation (DMR).

Parameter	IMR (n = 12) Mean ± SD	DMR (n = 22) Mean ± SD	*p*-Value
Age	65.0 ± 7.3	57.6 ± 11.7	0.058
Male, %	26.5	8.8	0.369
Weight	61.7 ± 8.2	67.8 ± 14.8	0.196
Height	1.6 ± 0.1	1.6 ± 0.1	0.986
BMI	23.3 ± 3.4	25.4 ± 4.0	0.136
BSA	1.7 ± 0.1	1.7 ± 0.2	0.301

BMI = body mass index; BSA= body surface area.

**Table 2 jcm-13-07503-t002:** Morphometric parameters of the mitral valve and left ventricle in ischemic and degenerative mitral regurgitation.

Parameter	IMR (n = 12) Mean ± SD	DMR (n = 22) Mean ± SD	*p*-Value
CC	37.9 ± 6.8	38.0 ± 8.9	0.978
TT	24.2 ± 3.4	22.6 ± 6.5	0.438
A_2_P_2_	9.0 ± 2.3	8.0 ± 2.7	0.293
Angle of tile of saddle	125.1 ± 10.3	121.7 ± 18.4	0.556
Area	12.4 ± 4.5	13.2 ± 5.9	0.664
Perimeter	13.0 ± 2.1	13.1 ± 3.0	0.890
AL	29.3 ± 7.3	30.3 ± 8.0	0.717
PL	15.7 ± 3.9	20.2 ± 5.6	0.018
Anterior coaptation	4.1 ± 1.0	4.0 ± 1.1	0.811
Posterior coaptation	4.1 ± 0.9	4.0 ± 1.1	0.837
Ch-A (s)	31.0 ± 5.4	32.2 ± 6.9	0.597
Ch-P (s)	25.3 ± 4.6	26.7 ± 6.9	0.549
Ch-A (d)	27.5 ± 7.8	27.8 ± 7.1	0.918
Ch-P (d)	25.0 ± 7.34	25.6 ± 6.01	0.802
Inter-papillary distance/mm	38.0 ± 6.7	31.8 ± 6.1	0.009
Indexed PM displacement	22.8 ± 3.7	18.4 ± 3.5	0.002
Anterolateral	1.0 ± 0.0	1.0 ± 0.0	0.000
Posteromedial	1.0 ± 0.0	1.0 ± 0.0	0.000
LVIDd/cm	5.3 ± 0.7	5.0 ± 0.7	0.186
LVIDs/cm	3.8 ± 0.9	3.5 ± 0.7	0.242
EF/%	47.9 ± 12.3	54.0 ± 9.2	0.113
P_max_	18.1 ± 26.9	16.8 ± 28.3	0.900
P_mean_	10.4 ± 18.2	9.7 ± 18.1	0.911

CC = commissure-to-commissure distance; TT = trigone-to-trigone distance; A_2_P_2_ = annulus height; AL = anterior leaflet height of the mitral valve; PL = posterior leaflet height of the mitral valve; Ch-A (s) = anterior chord length in systole; Ch-P (s) = posterior chord length in systole; Ch-A (d) = anterior chord length in diastole; Ch-P (d) = posterior chord length in diastole; indexed PM displacement = indexed papillary muscle displacement; LVIDd/cm = left ventricular end-diastolic internal diameter; LVIDs/cm = left ventricular end-systolic internal diameter; EF/% = ejection fraction; P_max_ = maximum pressure gradient across mitral valve; P_mean_ = mean pressure gradient across mitral valve.

## Data Availability

The data used in this investigation are available upon request. The study’s corresponding author can be contacted if anyone is interested in requesting access to the data.

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
