# Peer review of "Three-Dimensional Modelling of Indexed Papillary Muscle Displacement in Patients Requiring Mitral Valve Surgery Using Four-Dimensional Echocardiography Variables"

_jcm, 2024, doi:10.3390/jcm13247503_

Round 1
Reviewer 1 Report
Comments and Suggestions for Authors
The paper is well described in English language.
I'd suggest authors to specify in detail the types of "other mitral valve anomalies" otherwise it is impossible to give effective significance to the obtained results and consenquently to the paper, indipendently on the statistical significance.
As a matter of fact the anatomy of congenital mitral valve anomalies is so heterogeneous (for example left AV valve in AVSD vs arcade mitral valve vs double orifice vs isolated cleft vs MVP vs congenital mitral valve stenosis) and is mandatory to specify what types are included in the cohort. If this group is too much heterogeneous and not comparable to the group 1 I'd suggest to focalize only on the group 1 analysis.
Author Response
Reviewer's Comments:
The paper is well described in English language.
I'd suggest authors to specify in detail the types of "other mitral valve anomalies" otherwise it is impossible to give effective significance to the obtained results and consenquently to the paper, indipendently on the statistical significance.
As a matter of fact the anatomy of congenital mitral valve anomalies is so heterogeneous (for example left AV valve in AVSD vs arcade mitral valve vs double orifice vs isolated cleft vs MVP vs congenital mitral valve stenosis) and is mandatory to specify what types are included in the cohort. If this group is too much heterogeneous and not comparable to the group 1 I'd suggest to focalize only on the group 1 analysis.
Response:
Thank you very much for taking the time to review our manuscript. Please find the detailed responses below and the corresponding revision in the re-submitted files.
We carefully reviewed our database and matched the cohorts of Ischemic Mitral Regurgitation (IMR) and Degenerative Mitral Regurgitation (DMR) to ensure clarity and facilitate an effective comparison between the two groups. We appreciate your observation regarding the heterogeneity of mitral pathologies. In response, we have consistently referred to Group 1 as IMR and Group 2 as DMR throughout the manuscript for better alignment. Additionally, the exclusion criteria in the Methods section have been revised as follows:
“Patients were excluded if they had congenital mitral valve anomalies of heterogeneous pathologies, secondary mitral regurgitation, other valvular heart diseases, prior cardiac surgery, no 4D echocardiography performed, or missing data.”
Reviewer 2 Report
Comments and Suggestions for Authors
Three-dimensional Modelling of Indexed Papillary Muscles Displacement in Patients Requiring Mitral Valve Surgery Using Four-dimensional Echocardiography Variables
The manuscript performed a retrospective study assessing the efficacy of 4D echocardiography on determining papillary muscle displacement with either ischemic mitral regurgitation or other mitral valve pathologies. The software used (QLAB) was still undergoing development at the time of the study. Given the more recent 4D technology, this study could impact imaging for patients with ischemic mitral regurgitation and even broader implications beyond this scope in the foreseeable future.
1. Materials and Methods: Please add a flow chart for the population numbers. 142 patients and exclusion criteria were applied to 34 patients, divided into groups 1 and 2. See eg below –
2. Discussion: A comparison must be made between the results observed in this study using 4D echocardiography and results obtained in other studies using 3D echocardiography and cardiac resonance imaging. Also include a paragraph about how this technique can be used in future clinical applications beyond patients with IMR, and where this can be applied to the maximal benefit.
This manuscript is very well put together and requires moderate revisions.

Author Response
Reviewer's Comments:
The manuscript performed a retrospective study assessing the efficacy of 4D echocardiography on determining papillary muscle displacement with either ischemic mitral regurgitation or other mitral valve pathologies. The software used (QLAB) was still undergoing development at the time of the study. Given the more recent 4D technology, this study could impact imaging for patients with ischemic mitral regurgitation and even broader implications beyond this scope in the foreseeable future.
- Materials and Methods: Please add a flow chart for the population numbers. 142 patients and exclusion criteria were applied to 34 patients, divided into groups 1 and 2. See eg below –
- Discussion: A comparison must be made between the results observed in this study using 4D echocardiography and results obtained in other studies using 3D echocardiography and cardiac resonance imaging. Also include a paragraph about how this technique can be used in future clinical applications beyond patients with IMR, and where this can be applied to the maximal benefit.
This manuscript is very well put together and requires moderate revisions.
Responses:
- Thank you for your valuable comment. We have added a study flow diagram (Supplementary Figure 1) as per your suggestion.
- Thank you for your very timely and valuable comment. We recognize the importance of comparing the results from this study using 4D echocardiography with those from other studies employing 3D echocardiography and cardiac resonance imaging (MRI). Such comparisons would allow us to validate the findings from 4D echocardiography, as each imaging technique has its own strengths and limitations. Cross-validation with more established methods like 3D echocardiography and MRI would help ensure the reliability and accuracy of the results. Additionally, different imaging modalities may vary in clinical applications, resolution, and accuracy. However, given the retrospective nature of this study, a direct comparison between the modalities was beyond the scope of our analysis. The homogeneity of data across different techniques could not be guaranteed, and as such, we are unable to make such comparisons.
As per your next comment, the following paragraphs has been added to the discussion:
“We recognize the importance of comparing the results from this study using 4D echocardiography with those from other studies employing 3D echocardiography and cardiac resonance imaging (MRI). Such comparisons would allow us to validate the findings from 4D echocardiography, as each imaging technique has its own strengths and limitations. Cross-validation with more established methods like 3D echocardiography and MRI would help ensure the reliability and accuracy of the results. Additionally, different imaging modalities may vary in clinical applications, resolution, and accuracy. However, given the retrospective nature of this study, a direct comparison between the modalities was beyond the scope of our analysis.
Looking ahead, 4D echocardiography holds significant promise for a wide range of clinical applications beyond its current use in IMR. This technique can be particularly beneficial in the assessment and management of other cardiac conditions such as DMR, congenital heart defects, and valvular heart diseases. Its ability to provide dynamic, real-time visualization of heart function and valve motion can enhance preoperative planning, guide surgical interventions, and improve patient outcomes. Moreover, 4D echocardiography could be applied in monitoring the progression of heart failure, evaluating left ventricular function, and assessing post-surgical valve repair or replacement. As technology advances, the potential for 4D echocardiography to aid in the personalized management of various cardiac conditions will continue to grow, making it a powerful tool in both clinical practice and research.”
Round 2
Reviewer 2 Report
Comments and Suggestions for Authors
All comments have been addressed.
Author Response
Thank you for your remarks. We are thankful for your valuable time to review our manuscript.